# Security or Safety: Quantitative and Comparative Analysis of Usage in Research Works Published in 2004–2019

**DOI:** 10.3390/bs9120146

**Published:** 2019-12-09

**Authors:** Olesia V. Bubnovskaia, Vitalina V. Leonidova, Alexandra V. Lysova

**Affiliations:** 1Department of Psychology and Education, School of Arts and Humanities, Far Eastern Federal University, 8 Sukhanova St., 690091 Vladivostok, Russia; 2Department of Philosophy and Religion Studies, School of Arts and Humanities, Far Eastern Federal University, 8 Sukhanova St., 690091 Vladivostok, Russia; leonidova.vv@dvfu.ru; 3School of Criminology, Simon Fraser University, Vancouver, BC V6B5K3, Canada

**Keywords:** psychology, safety, security, psychological security, psychological safety, human well-being

## Abstract

This article is devoted to the statistical analysis of security and safety frequency in the context of categories connected with social institutions and personality features in research works from 2004–2019. Research was based on the following methods: quantitative analysis of safety frequency in the context with coded “categories” related to social institutions and personality features; analysis was conducted with computer-assisted content analysis QDA Miner Lite v. 1.4 and Fisher’s F-test. An analysis of 1157 works showed that the terms “security” and “safety” were quantitatively more frequent when used with concepts related to social institutions than with concepts related to personality features. In our opinion, this qualitative trend shows the prevailing significance of social aspects of security over its personal (psychological) traits for research analysis and practical social aspects. The priority usage of the terms “security” and “safety” can be related to the securitization of society, (i.e., to the increased role and significance of social ways of providing security and protection from threats), primarily with the help of external law-enforcing actors such as the state, police, and army. Securitization counterweights the development of social and psychological mechanisms of security—developing motivation for safe behavior, personal self-regulation, and self-production of security as an internal feeling of protection.

## 1. Introduction

The concept of security in science is presented in two major ways and modes: safety and security. The Oxford Dictionary suggests the difference between these two concepts in the element of protection: safety involves creating protection from risks or dangers whereas security means the state of being free from danger or threats [1,2].

The first major statement concerning human security appeared in the 1994 Human Development Report (HDR), an annual publication of the United Nations Development Programme (UNDP). The definition was comprehensive and included security from such chronic threats as hunger, disease, and repression and, at the same time, protection from sudden and hurtful disruptions in the patterns of daily life, whether in homes, jobs or communities. Although the UNDP’s 1994 definition of human security remains the most widely cited and ‘most authoritative’ formulation of the term, some countries, such as Canada, use a more restrictive definition of human security as “freedom from pervasive threats to people’s rights, safety or lives.” There were other attempts as well to narrow the concept, including one that identified the key indicators of well-being—poverty, health, education, political freedom, and democracy [3].

Another line in the evolution of a human security concept limited compartmentalization and focused on personal security [4]. One of the elements of personal security is psychological security, not explicitly discussed in the 1994 HDR. Yet, it is fundamental in life experiences, and central to peace and human dignity and a basis for effective personal agency. The psychological aspects of ‘personal security’ are crucial elements in human security research and policy agenda [4]. At the same time, in the new Routledge Handbook of Human Security [5], the term ‘human security’ occurs over 2400 times, but ‘personal security’ is found only three times. In comparison, other frequencies are as follows: food security 20×; environmental security 8×; economic security 6×; health security 6×; political security 6×; and community security 2×. Some comparable terms rank as follows: national security 136×; state security 33×; global security 16×; and military security 8×.

Modern researchers [6,7] who analyze the occurrence of safety and security terms point out the commonality of these concepts in everyday speech [7] (p. 329); both terms stand for the state or conditions of the absence of threats and dangers but are differentiated in their subjects. Safety often implies the tools that provide for protection; security is about the actors [7] (p. 329). However, comprehension of these two terms in the academic discourse is quite challenging [7] (pp. 321–322), [8]. On the one hand, the differences between safety and security are actively conceptualized; on the other hand, they lack unambiguous definitions, particularly since researchers have yet to decide if they should include the everyday use of these terms in their definitions [7] (p. 322).

Meanwhile, sociological, psychological, and anthropological research of security [9,10,11,12] show the currently prevailing trend of life securitization that involves the increased significance of external actors ensuring the security in the context of various threats and protection against them [9] (p. 282). Under these conditions, security is understood and analyzed as a condition connected with the forces and processes of preserving and maintaining order in the society (security), thus being different from the personal perception of security (safety) [11].

Therefore, our research is to find out if the trend of society securitization has been reflected in the academic discourse of security. In order to confirm this assumption we studied the development of research analysis of these concepts.

## 2. Materials and Methods

In order to substantiate the trend of societal securitization and securitization of the human security understanding, we studied how the academic understanding of these concepts has evolved.

We selected and analyzed the research works from Scopus, a major international citation database.

First, we performed a statistical analysis of the works reviewing security-related topics. Works published in the last 15 years were selected and broken up into three five-year periods: 2004–2009, 2010–2014, and 2015–2019. This 15-year period was chosen as the number of security-related works has more than doubled since 2004 thus ensuring the sampling reliability and research relevance. It has yet to be taken into consideration that the year 2019 has not been studied in the full volume.

A simple search method was applied to select articles from the Scopus database upon the criterion of ‘title of the article, short description, and keywords’ limited by the above-mentioned periods; it resulted in the following number of articles: 293,331, 373,844, and 380,369, respectively. After the criterion of objectness was included into the search parameters (social science, arts and humanities, psychology, neuroscience) we obtained the outcome of 21,413, 32,123, and 36,489 articles. The filter “sort by relevance” was then used to arrange the search results. Special attention was paid to the titles, keywords, and extracts in the article sampling of each period. As a result, 1157 articles were selected: 310 in 2004–2009, 359 in 2010–2014, and 488 in 2015–2019.

Every sample group was analyzed with computer-assisted qualitative analysis software QDA Miner Lite v. 1.4. Statistical analysis required the coding of several word groups. Firstly, we coded the concept of security as ‘safety’ and ‘security’ in order to analyze their possible connections with other codes and put them into a group referred to as ‘general’.

Secondly, we coded different aspects of a social system and personality that can be related to the problem of security and formed the following groups:-social institutes, including the concepts such as ‘state’, ‘education’, ‘family’, ‘military’, ‘economy’, ‘religion’;-threats (‘terrorism’, ‘disease’, ‘wars’, ‘crime’, ‘violence’), included into the semantic field of such concepts as ‘safety, security, risk’ [6] (p. 209) and create the phenomena that endanger security;-gender (‘male’, ‘female’, ‘LGBT (LGBTIQ+)’, ‘queer’);-age (‘children’, ‘young’, ‘teenagers’, ‘adults’, ‘elderly’);-needs (‘food’, ‘health’, ‘sex’, well-being’, ‘culture’);-minority (‘migrants’, refugees’, disabled people’);-types, (‘psychological’, ‘personal’, ‘social’).

Each concept from these groups was a separate code that coded a related part of the article. The whole articles (a case) as units to be coded were selected for the analysis. The code was assigned if a coded concept (a code) was revealed in the cases that were being studied.

## 3. Results

The process of coding and subsequent extraction of codes from the security-related articles brought the following results: the codes that were most frequently used within that period of time were ‘health’, ‘state’, and ‘education’. The codes ‘children’, ‘family’, ‘culture’, ‘female’, ‘male’, ‘social’, and ‘personal’ were less frequent, however, also quite relevant. These codes fell into four of the seven groups we previously defined: ‘social institutes’ (the highest number of relevant codes), ‘gender’, needs, and types.

A more detailed statistical analysis of articles based on these codes shows:

1. Statistics of code frequency in articles (except for safety and security).

All three samples of publications (Figure 1, Figure 2 and Figure 3) most frequently contain the codes “education”, “state”, and “health”. The figures below show the frequency for each code in different timeframes.

In 2004–2009 (Figure 1) the code “education” was found 332 times, the code “state” was found 340 times, and the code “health” was found 212 times.

In 2010–2014 (Figure 2) the code “education” was found 427 times, the code “state” was found 384 times, and the code “health” was found 264 times.

In 2015–2019 (Figure 3) the code “education” was found 668 times, the code “state” was found 620 times, and the code “health” was found 384 times.

2. Statistics of case frequency (percentage of cases) with the codes and of the codes (percentage of codes).

The cases with codes “health”, “education”, and “state” prevailed in terms of the quantity of works where the codes occured.

In 2004–2009 (Figure 4) the frequency of cases with codes ”health” and “state” comprised 68.7% of works; the frequency of code “education” was 62.6% of works.

In 2010–2014 (Figure 5) the frequency of cases with the code “health” comprised 73.5% of works; the frequency of code “education” was 69.9% of works; and the frequency of code “state” was 65.50%.

In 2015–2019 (Figure 6) the frequency of cases with code “health” was almost 80% of documents; the frequency of code “education” was 74% of documents; the frequency of code “state” was 79.9%.

The codes “education”, “state” and “health” prevailed in terms of the code frequency in the works.

In 2004–2009 (Figure 7) the code “education” covered 11.4% of all codes found in the works; the code “state” covered 11.7%; and the code “health” covered 7.3%.

In 2010–2014 (Figure 8) the code “education” covered 11.9% of all codes found in the works; the code “state” covered 10.6%; and the code “health” covered 7.3%.

In 2015–2019 (Figure 9) the code “education” covered 11.9% of all codes found in the works; the code “state” covered 11.1%; and the code “health” covered 6.8%.

We also compared the usage of ‘safety’ and ‘security’ in the context with the most relevant codes by applying coding retrieval and retrieving the codes ‘safety’ and ‘security’ in sequence on the condition of enclosing (when the target code is amid the preselected codes).

The code ‘safety’ is most frequently found in context with the following codes (Figure 10): ‘health’—213 cases in the sampling of 2004–2009, 264 cases in the sampling of 2010–2014, 385 cases in the sampling of 2015–2019; ‘state’—202 in the sampling of 2004–2009, 235 cases in the sampling of 2010–2014, 391 cases in the sampling of 2015–2019; ‘education’—193 in the sampling of 2004–2009, 250 cases in the sampling of 2010–2014, 362 in the sampling of 2015–2019.

Frequency analysis has shown the prevailing significance of the code ‘education’ whereas here it ranked third, having been surpassed by the code ‘health’ which ranked first.

The code ‘security’ is most frequent in the context with ‘state’ (75, 80, and 155 cases), with ‘health’ (64, 67, and 135 cases), and ‘education’ (61, 70 and 131 cases), respectively (Figure 11). Thus, the code ‘state’ has the highest significance in the works devoted to security issues.

Comparative analysis of occurrence frequency of these terms based on the objectness criterion was conducted in psychology articles. Throughout the whole period including 2004–2009, 2010–2014, and 2015–2019 the frequency of ‘security’ in psychology was slightly higher in comparison with ‘safety’ (1.5% and 1.1%, respectively).

The Fisher’s F-test was used to reveal the statistically significant differences in occurrences of psychological safety and personal safety (8.9% and 2.7%, respectively), as well as ‘psychological security’ and ‘personal security’ (18.5% and 1.9%, respectively) in articles on psychology (φ = 1.952, *p* < 0.05 and φ = 4.321, *p* < 0.01, respectively).

Analysis of code combinations has shown that «psychological security» was most frequent in psychology-related contexts in comparison with ‘psychological safety» (18.5% and 8.9%, respectively, φ = 2.001, *p* < 0.05); analysis of the personality-related context did not show any statistically significant differences between ‘personal safety’ and ‘personal security’ (2.7% and 1.9%, respectively, φ = 0.375).

## 4. Discussion

The highest relevance of the “state” category in research publications can confirm the assumption about “securitization” of the safety discourse. In the anthropological research that set forward the idea of securitization [11], security involves maintaining the regulatory order in society via producing various threats and threat protection [9] (p. 282) by government order and control [10] (p. 487), whereas key security issues include health threats, safety of the urban environment, production of fears, criminalized groups, migrant threats, terrorism and extremism, forces and institutes of security [11,12]. However, in the light of their usage both with “security” and “safety”, the categories “health” and “education” also confirm the assumption about securitization.

Analysis of the article content with the relevant categories “state”, “health”, and “education” shows that these publications are devoted to political (organizational and administrative), and police measures in the area of public security connected with health protection [13,14,15,16,17,18,19,20,21,22,23,24,25]; food security [26,27,28]; protection from threats and violence in educational sphere [29,30,31,32,33,34,35,36,37,38,39,40]; urban security [41,42,43,44,45,46,47]; and protection from crime and terrorism [48,49,50,51,52,53,54,55]. These life spheres and human behavior are evaluated for external risks and physical threats with preventive or reactive impacts on the environment and social relations being the factors and tools of protection from them.

Despite the relatively low relevance of the category “safety” when used in personal and psychological contexts, this term shows higher frequency in comparison with the term “security” when used in the contexts with codes “health”, “education”, and “state”. The highest frequency of the code “health” in this situation confirms the priority of discussing personal safety primarily in the context of health issues that have been confirmed by the works studied. However, it is again about physical protection of health from external threats (food, medical, industrial).

Only a few works [56,57,58,59,60,61] are devoted not only to safety and its perception connected with external protection from threats but also to the internal feeling of safety, its factors, and tools of self-production.

## 5. Conclusions

Statistical analysis of research works published in 2004–2019 aimed at security-related terms has confirmed the idea of securitization of human security expressed in the anthropological literature. This trend is mostly seen in the prevailing significance of social institutions (external actors) over personal (psychological) aspects of security in the context of its research. The content analysis of sampled articles has confirmed the conclusion about securitization of safety discourse since they are mostly devoted to social (political) practices of security management in various life spheres. The articles that study psychological (personal) aspects of security were less numerous in the sampling; however, they also set forth the important question of insufficient involvement of perceptive components in security studies.

Research limitations arise from the fact that the articles for analysis have been taken from the academic areas that mostly study humanitarian and social problems of security. The future studies will cover the analytical materials that will include the articles in engineering and computer science; these materials will add more insight into the concept of security through studying its aspects in technological systems.

Results of our research can be used in developing security programs in social and educational spheres as well as in research of security perception issues.

## Figures and Tables

**Figure 1 behavsci-09-00146-f001:**
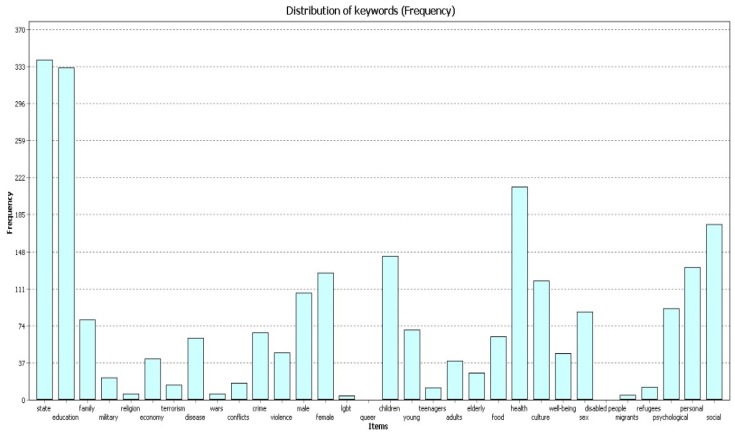
Frequency, 2004–2009.

**Figure 2 behavsci-09-00146-f002:**
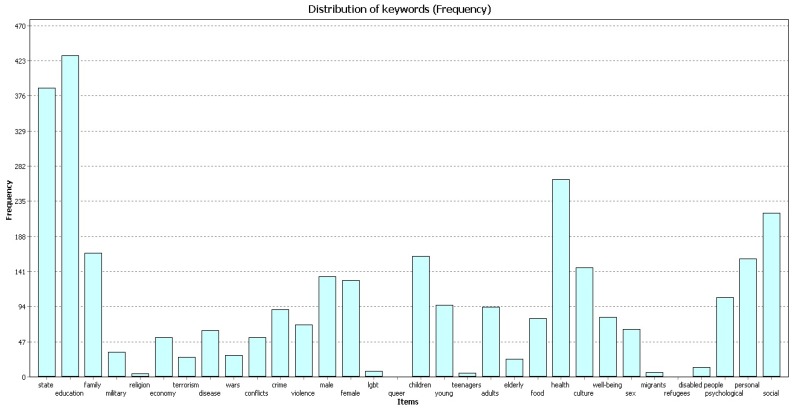
Frequency, 2010–2014.

**Figure 3 behavsci-09-00146-f003:**
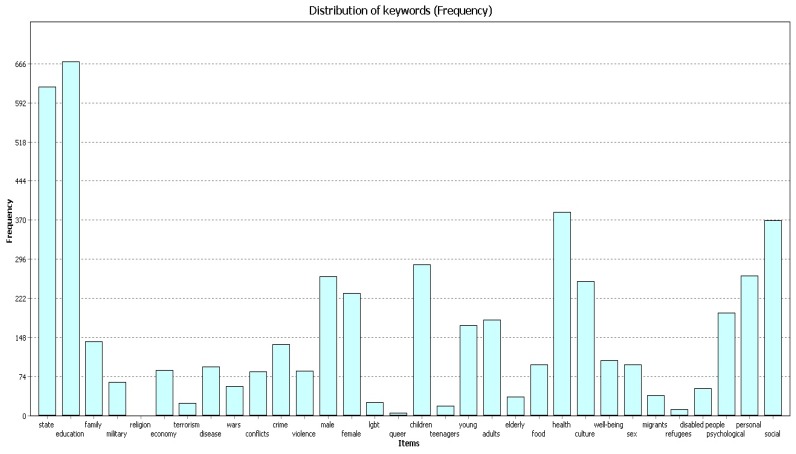
Frequency, 2015–2019.

**Figure 4 behavsci-09-00146-f004:**
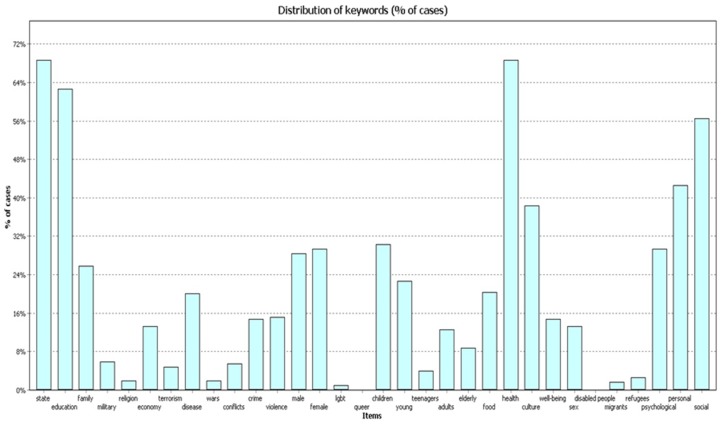
Percentage (%) of cases, 2004–2009.

**Figure 5 behavsci-09-00146-f005:**
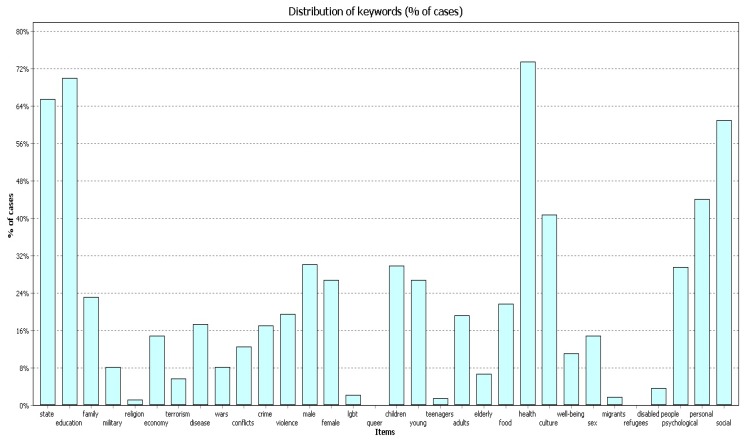
Percentage (%) of cases, 2010–2014.

**Figure 6 behavsci-09-00146-f006:**
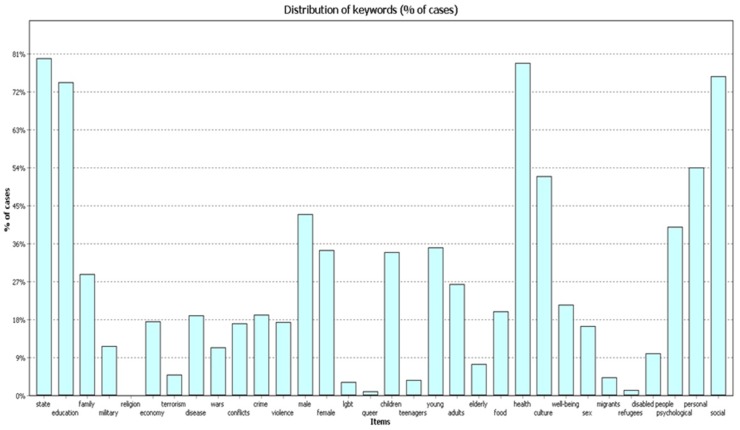
Percentage (%) of cases, 2015–2019.

**Figure 7 behavsci-09-00146-f007:**
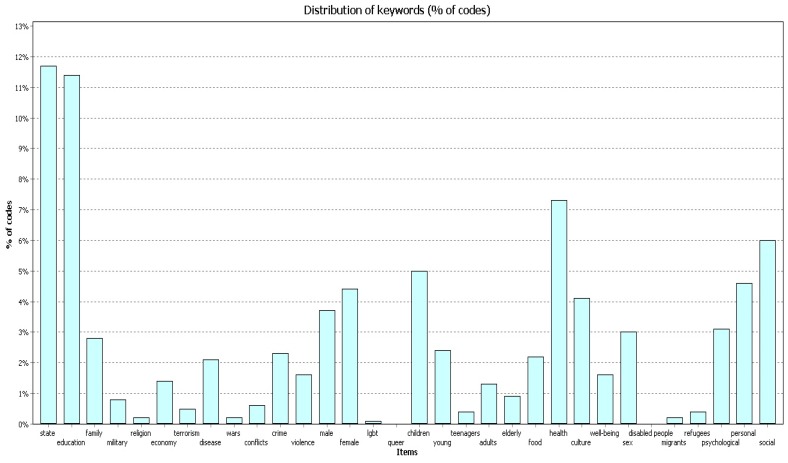
Percentage (%) of codes, 2004–2009.

**Figure 8 behavsci-09-00146-f008:**
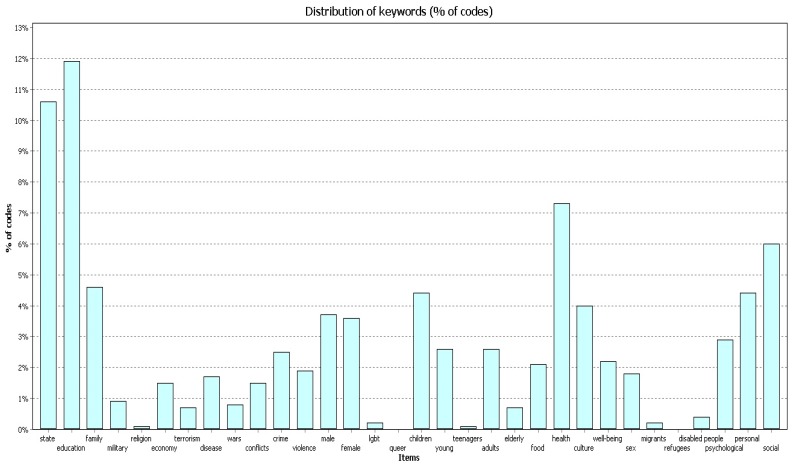
Percentage (%) of codes, 2010–2014.

**Figure 9 behavsci-09-00146-f009:**
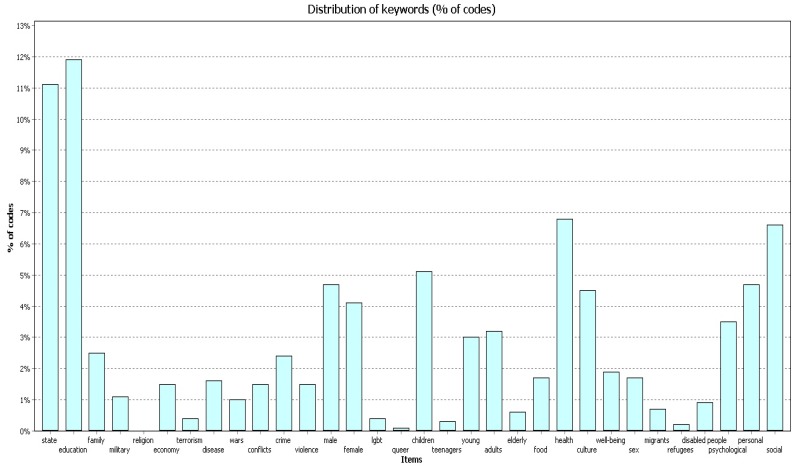
Percentage (%) of codes, 2015–2019.

**Figure 10 behavsci-09-00146-f010:**
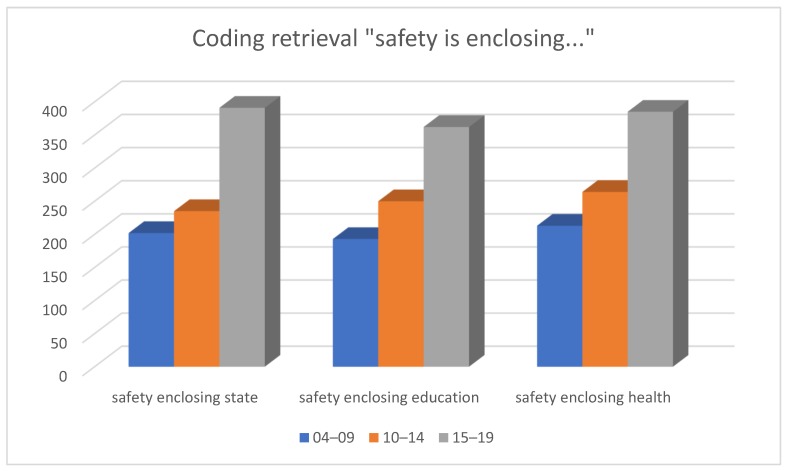
Results of coding retrieval, ‘safety’.

**Figure 11 behavsci-09-00146-f011:**
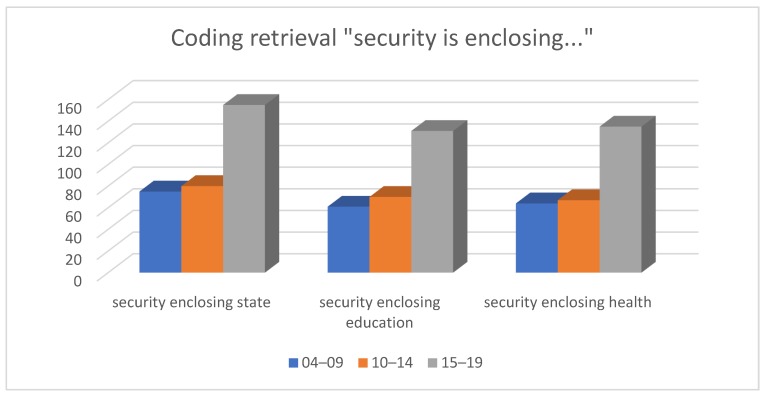
Results of coding retrieval, ‘security’.

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
