# Peer review of "Security or Safety: Quantitative and Comparative Analysis of Usage in Research Works Published in 2004–2019"

_behavsci, 2019, doi:10.3390/bs9120146_

Round 1

Reviewer 1 Report

I am content with the changes effected. Article requires moderate English language checks

Author Response

reviewers' comments

respond

Article requires moderate English language checks

The manuscript was checked carefully by native speaker, he corrected the text.

Also he noted long sentences are difficult to perceive, but it shouldn’t change, because it will distort the meaning and break the author's style

Reviewer 2 Report

I have carefully reviewed the modifications introduced by the authors in relation to the suggestions indicated in the first review. The authors have responded satisfactorily to some of the questions requested, but there are some relevant aspects that have not been addressed.
Method:
Lines 83-84: the authors still do not clarify what has been the "relevance criterion" adopted to obtain the 1157 articles reviewed for research.
Conclusions:
The practical implications of the results obtained remain unknown. The authors simply describe the results. On the other hand, neither the limitations of the study nor the future lines of research have been indicated.

Author Response

reviewers' comments

respond

English language and style are fine/minor spell check required

The manuscript was checked carefully by native speaker, he corrected the text.

Method:

Lines 83-84: the authors still do not clarify what has been the "relevance criterion" adopted to obtain the 1157 articles reviewed for research.

We described how we used “relevance criterion” in Scopus

Conclusions:

The practical implications of the results obtained remain unknown. The authors simply describe the results. On the other hand, neither the limitations of the study nor the future lines of research have been indicated.

We described the practical implications of the results, the limitations of the study and future line of research

This manuscript is a resubmission of an earlier submission. The following is a list of the peer review reports and author responses from that submission.

Round 1

Reviewer 1 Report

Title - The title of the paper is not clear enough – The authors should restructure the title to be clear enough for readers.

Line 22: Authors should provide citation for the statement.

Line 23 – 24: The statement seems vague. Authors should provide information about the indices. A few lines about that will suffice.

The authors failed to fail to point out the contribution of their study into the body of knowledge succinctly.

The paper requires overhauling in terms of language use. Also, discussion and presentation of the results should be strengthened.

Finally, the conclusion of the study need to be strengthened. The authors should discuss the implications of their findings.

Reviewer 2 Report

Thank you for the opportunity you have given me to review the manuscript. In my opinion, the work presented by the authors shows several weaknesses in scope that lead me to discourage the publication. These weaknesses are as follows:

Abstract:

It is too concise and lacks the recommended structure in a psychological study (follow the Introduction, Methods, Results, Discussion, IMRD format). It would be necessary to introduce some information that serves as a conceptual frame of reference. Likewise, the number of studies that were reviewed should be included. There is also an absence of some kind of conclusion regarding the results obtained.

Introduction:

Lines 64-65: What is the relationship between this question and the investigation? In my opinion, the relevance and contribution of the study are not justified in a clear way.

Material and methods:

Lines 67-68: This phrase seems to be the objective of the study, so it should be located in the Introduction. However, the reason for the review is not justified, and no research hypothesis is formulated. What reason has motivated such an increase in the number of security publications since 2004? Until what month of 2019 was the study planned? Line 79: “The criteria of relevance…” What criteria is this? The criteria adopted to determine the 1157 papers that were reviewed for this study is not properly justified. Lines 99-103: this paragraph describes results, so it should go in the Results section.

Conclusions:

This section is very poor. It requires much more elaboration. A more precise and in-depth analysis of the results obtained and a true discussion of them are required. Likewise, the implications of the results must be indicated. Nor are the limitations of the study or future lines of research specified.

References:

Further deepening of the sources consulted is required. According to the authors, there is a lot of research on the analyzed topic, which contrasts with the scarce 13 references used in this study as documentation.

Reviewer 3 Report

Line No.

65 - state being safe instead of protected.

78 - disciplines are proper nouns and should be in lower case.

Each figure would be introduced separately before the figure and each should be summarized separately after the figure. Figures are different, so treat them separately.

in the figure headings, write percent in stead of %.  

120-122 - The symbol (%) is used within tables and figures, but not in text.  Write percent in the text as well.